# Movement-Based Participatory Inquiry: The Multi-Voiced Story of the Survivors Justice Project

Kathy Boudin [1], Judith Clark [2], Michelle Fine [3,*], Elizabeth Isaacs [4], Michelle Daniel Jones [5], Melissa Mahabir [6], Kate Mogulescu [4], Anisah Sabur-Mumin [6], Patrice Smith [6], Monica Szlekovics [6], María Elena Torre [3], Sharon White-Harrigan [7] and Cheryl Wilkins [1]

1 Center for Justice, Columbia University, New York, NY 10027, USA; kb2023@columbia.edu (K.B.); cw2555@columbia.edu (C.W.)
2 Hour Children, New York, NY 11106, USA; judynyc49@gmail.com
3 Public Science Project, The Graduate Center, City University of New York, New York, NY 10016, USA; mtorre@gc.cuny.edu
4 Brooklyn Law School, Brooklyn, NY 11201, USA; elizabeth.isaacs@brooklaw.edu (E.I.); kate.mogulescu@brooklaw.edu (K.M.)
5 Constructing Our Future, New York University, New York, NY 10012, USA; mcj320@nyu.edu
6 Survivors Justice Project, Brooklyn, NY 11201, USA; melissa119@gmail.com (M.M.); anisah613@gmail.com (A.S.-M.); patricesmith31@icloud.com (P.S.); monicaszlekovics@gmail.com (M.S.)
7 Women Community Justice Association, Brooklyn, NY 11208, USA; sharon@wcja.org
* Correspondence: mfine@gc.cuny.edu

**Abstract:** We write as the Survivors Justice Project (SJP), a legal/organizing/social work/research collective born in the aftermath of the 2019 passage of the New York State Domestic Violence Survivors Justice Act (DVSJA), a law that allows judges to re-sentence survivors of domestic violence currently in prison and to grant shorter terms or program alternatives to survivors upon their initial sentencing. Our work braids litigation, social research, advocacy, organizing, popular education, professional development for the legal and social work communities, and support for women in prison going through the DVSJA process and those recently released. We are organized to theorize and co-produce new knowledges about the gendered and racialized violence of the carceral state and, more specifically, to support women currently serving time in New York State to access/understand the law, submit petitions, and hopefully be freed. In this article we review our collective work engaged through research and action, bridging higher education and movements for decarceration through racial/gender/economic justice, and venture into three aspects of our praxis: epistemic justice in our internal dynamics; accountabilities and deep commitments to women still incarcerated and those recently released, even and especially during COVID-19; and delicate solidarities, exploring external relations with policy makers, judges, defense attorneys, advocates, and prosecutors in New York State, other states, and internationally.

**Keywords:** domestic violence; decarceration; critical participatory action research; solidarity

## 1. Introduction

A law is only as powerful as the legal advocacy, relentless organizing, and community-based inquiry that comes before and comes after. This is the story of a university–community collaborative that developed across institutions, community organizations, and nodes of feminist and racial activism to facilitate implementation of the Domestic Violence Survivors Justice Act (DVSJA).

We write as the Survivors Justice Project (SJP), a legal/organizing/social work/research collective born in the aftermath of the 2019 passage of the New York State Domestic Violence Survivors Justice Act. With many caveats (elaborated below), the DVSJA allows judges to re-sentence survivors of domestic violence currently in prison and to grant shorter

terms or program alternatives to survivors upon their initial sentencing. The passage of the law was the result of a 10-year struggle launched and led by survivors of both incarceration and domestic violence, largely women of color. Our work braids litigation, social research, advocacy, organizing, popular education, professional development for the legal and social work communities, and support for women in prison going through the DVSJA process and those recently released. We are organized to theorize and co-produce new knowledges about the gendered and racialized violence of the carceral state and, more specifically, to support women currently serving time in New York State to access/understand the law, submit petitions, and hopefully be freed. We also conduct trainings with judges, prosecutors, and defense attorneys; seek to make visible the intimate relationship between domestic violence and the criminalization of women, especially women of color, for journalists and the general public; and collaborate with interdisciplinary research collectives across New York, in other states, and internationally. We model for higher education critical participatory research and action, centering the perspectives and knowledge of those most impacted.

We are a collective of lawyers, advocates, organizers, researchers, and social workers; most of us have survived prison and/or domestic violence. We came together in 2020, just after the passage of the DVSJA, to document, litigate, educate, and build community with women in prison and those recently released. Our focus has been on the part of the law that enables survivors currently in prison to apply for re-sentencing.

We knew well that the power of this law would only be as strong as what would follow: the legal petitions, research, organizing, popular education, and, importantly, survivors' ability to safely process and share their narratives of trauma and abuse. Together we represent a range of "partner" organizations across the state and within New York City. In the spirit of solidarities grown over time, many of us have been working together since 1994, when we collaboratively documented the wide-ranging impact of college in prison and published "Changing Minds: The Impact of College in Prison on Women, their Children, the Prison Environment and Post-Release Outcomes" (Fine, Torre, Boudin, Bowen, Clark, Hylton, Martinez, Roberts, Smart, Upegui, Wilkins); many of us lived together for decades or years at Bedford Hills Correctional Facility, one of three prisons in New York for those identified by the state as women; some of us were very recently released, and some of us have never been incarcerated.

In this article, we review our collective work engaged through research and action, bridging higher education and movements for decarceration through racial/gender/economic justice, while accounting for geographic differences. We venture into three aspects of our praxis. First, we explore our commitment to epistemic justice in our internal dynamics: how we co-produce knowledge with critical sensibilities and ethics (Alcoff 2018; Anzaldúa 2003; De Sousa Santos 2014; Fine and Ruglis 2009; Fine and Torre 2021; Fine et al. 2021; Torre and Ayala 2009; Tuhiwai Smith 1999), how we negotiate what Gloria Anzaldúa (1987) called "choques"/tensions/power dynamics among us, and how we navigate our resources and differences, by discipline and training, generations, varied experiences in prison and with domestic violence, race/class, and region. We center those with experiences of surviving domestic violence and prison, who have—as significantly—spent substantial time talking and organizing with peers and comrades, working for social change, who introduced experience and a crucial line of analysis rooted in struggle. Second, we reflect on our accountabilities and deep commitments to women still incarcerated and those recently released, even and especially during COVID-19. You will hear our attempts to identify women in prison vulnerable to COVID-19 so that we might submit their names to the governor for early release; our conversations with women inside to help design a "resource guide" for navigating the DVSJA process (extremely powerful); and our discussions with women recently out about the joy and disappointments of "freedom" into a deeply unjust world (beautiful and heart-breaking). Third, we introduce scenes of delicate solidarities, exploring external relations with policy makers, judges, defense attorneys, advocates, and prosecutors in New York State, other states, and internationally. By peering into these three

contexts, we animate the joys, magic, and struggles of critical collaborative inquiry/action, dedicated to decarceration and against domestic violence, through a feminist and critical race lens. As you read, you will hear our collective voice braided with individual reflections noted by our first names.

## 2. The Shape of This Article

This article is not structured as a typical research-based document. Instead, we offer a detailed image for readers of an activist *research-policy-action project*, undertaken by an interdisciplinary team, led by the insights and incites of women who have been incarcerated and those who have survived domestic violence. We do not present research questions or findings as such, but we sketch a praxis for interrogating, understanding, reframing, decolonizing, and litigating against the tight braid of domestic violence and the criminalization of women, particularly women of color. This paper offers an epistemic journey in-to our project and then out-toward the many policy makers and social movements with whom we collaborate. As a research collective of lawyers, social scientists, activists and organizers, rooted in the wounds and knowledge borne in domestic violence and behind bars, we position intimate, racial, and state violence against women to be an instance of what Massey and Hall (2010) would call *conjunctural crises*—where seemingly autonomous social forces combine to disproportionately dispossess highly marginalized communities. In such struggles, critical, participatory inquiries by and for those most impacted are crucial for policy making, community organizing, popular education, and theory building. We offer here a deep reflective consideration of one such research collective—Survivors Justice Project—dedicated to social transformation, theory building, and to be of use to racialized communities under siege and feminist movements for decarceration and abolition.

## 3. In a Legacy of Struggle Led by Women in and out of Prison

Like all movements for justice, the Survivors Justice Project has been nourished by a long legacy of struggle led by women in and out of prison and those impacted by the criminal punishment system. Across this brief timeline, you will hear private and then public gatherings of women, often survivors of state violence and intimate violence, gathering, sharing stories, speaking aloud, demanding justice. This is the her-story of racial and gendered justice. It is a testimony of the shoulders SJP sits on, the long line of women in prison and out, largely women of color, who have struggled for decades to challenge the enduring linkages between domestic violence and the criminalization and incarceration of women. Our work connects back 35 years to a public hearing on domestic violence and criminalization of survivors held in 1985 at Bedford Hills Correctional Facility, New York's maximum security women's prison. The hearing was initiated by women at Bedford and held within the facility with allies and advocates in state government and community and policy makers in Albany who sought to document how systematically violence at home was associated with the criminalizing of survivors.

Women testified about how their experiences of domestic violence led to their criminal legal involvement. They proposed ideas on how to respond to the ongoing punishment of women who have survived violence.

These testimonies stand as evidence of women's carceral wisdom (McGinty Forthcoming) borne in collective intimate spaces, shattering the silences that (continue to) surround domestic violence and its relation to women's incarceration. The hearing was the culmination of months of small, quiet conversations that transformed into consciousness-raising among a most diverse group of women imprisoned in Bedford Hills—women from urban New York City, Syracuse and Buffalo, Suffolk County, Long Island, and small towns across New York State; women who identified as poor/working class and wealthy; White, Black, Latina, Asian American, and mixed race women; women who were beaten, emotionally abused, and/or sexually victimized by mothers/fathers/stepfathers/male partners and other women. Advocates in the governor's office, as well as community-based activists, worked with the women over time to curate new narratives that would integrate the vio-

lence they experienced in their homes with the systemic sexism, racism, neglect, and abuse they encountered from public agencies tasked with "helping" families. Women shared shockingly resonant—and trans-historic—stories about police "walking him around the block", Child Protective Services "threatening to take my children", lawyers who recommended "that I not mention the violence as part of my defense" (Bedford Hills Hearings transcript 1985 and video provided by (Eldridge 2021)), and judges who would ultimately impose punishments for their attempts to survive. In quiet sessions with peers and then advocates and then aloud in the gymnasium, women narrated their traumas. This was a moment when, in the language of Mills (1959), a "private" issue, silenced and trivialized, emerged as a public concern—and a political struggle—for women, across race/class and community.

In 2009, as part of the continued organizing by and for women across New York State, the Coalition for Women Prisoners launched the Domestic Violence Survivors Justice Act advocacy campaign. Their goal was to dismantle the relentless, racialized, and gendered system of criminalization and punishment with legislation that would expand judicial discretion to offer survivors less punitive sentencing. Currently and formerly incarcerated women, many of whom were survivors of domestic abuse, led the campaign in partnership with domestic violence advocates, sentencing reform experts, attorneys, judges, criminal reform organizations, women's organizations, and community leaders. Ten years after the campaign's launch, the New York State legislature passed the DVSJA in 2019 (Figure 1).

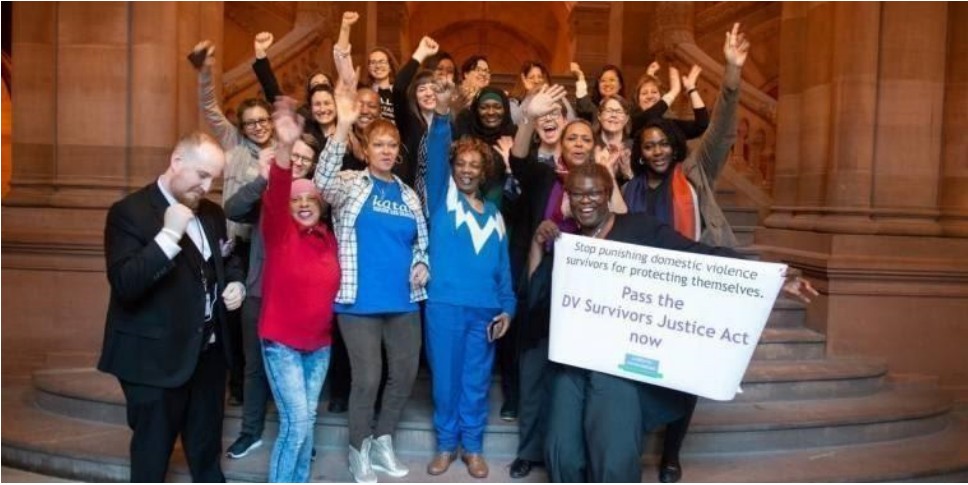

**Figure 1.** Lobbying for passage of the DV Survivors Justice Act (Photo by Nasir Abdul-Mumin).

Both Anisah Sabur-Mumin and Sharon White-Harrigan were instrumental in the development and passage of the law.

Anisah remembers the momentum of the struggle:

"Being able to organize with women who experienced the criminalization and victimization of abuse and the legal system was one of most powerful and enlightening experiences of my lifetime."

Then, as now, the opportunity to work across, with women inside and out, was key to solidarity. As Anisah describes,

"Leading the organizing for the last four years of the campaign gave me great pleasure to stand alongside my sisters inside and outside to become victorious in this campaign."

The law allows a court to impose an alternative sentence, or to re-sentence someone already in prison, if it finds that: (1) a person was, at the time of the offense, a victim of domestic violence subjected to substantial physical, sexual, or psychological abuse inflicted by an intimate partner or family member; (2) the abuse was a "significant contributing factor" to their participation in the crime; and (3) a sentence under current law would be, or in the case of re-sentencing the original sentence is, "unduly harsh". If the court finds that these criteria are met, the law allows judges to sentence survivors to shorter prison

terms and, in some cases, community-based alternative-to-incarceration programs and to reduce the sentences of survivors currently in prison.

While these criteria reflect compromises negotiated with state legislators, the DVSJA nonetheless retains important precedent-setting elements. The law takes a broad view of domestic violence as more than just physical abuse to acknowledge the impact of sexual abuse and psychological abuse/coercive control. It also recognizes that domestic violence is not limited to intimate partner relationships and includes abuse committed by other family members or guardians. Most significantly, the DVSJA allows relief for survivors convicted of a range of offenses, including felonies classified as "violent". The law does not specify who the harmed person must be; relief is available to survivors convicted of crimes against their abusive partner as well as crimes against strangers and others, if the survivor can provide evidence that s/he/they was a survivor of domestic violence at the time of the offense and show a connection between the abuse and the offense itself. In sum, the DVSJA incorporates a comprehensive understanding of how violence and coercion at home can affect survivors.

## 4. SJP: A Participatory Research/Action Collective Rooted in the Wisdom of Survivors and Accountable to Those Behind Bars

When the law went into effect, the New York State Department of Corrections & Community Supervision (DOCCS) provided lists of everyone in custody at the time who was potentially eligible for re-sentencing under the DVSJA based solely on charge and sentence. In addition to a list of people in men's prisons, the list contained the names of the 487 people in women's facilities (DOCCS 487) who were sentenced before the DVSJA went into effect (as anyone sentenced after the law passed is only able to obtain DVSJA relief at their initial sentencing), were serving eight or more years, and were not convicted of one of the excluded offenses. We began with a collective knowledge building session, inviting law students, critical psychology doctoral students, and gender/women's studies students, to a teach-in by Judy Clark, Sharon White-Harrigan, Cheryl Wilkins, and Michelle Daniel Jones on how domestic violence lives within prisons; how the DVSJA might activate possibilities for women inside; how domestic violence, trauma, and mental health are addressed behind bars; how women inside offer each other mutual aid and DV consciousness raising; how news and details of the DVSJA might circulate within prisons; and how our research might be of use. At that meeting we began to build SJP: lawyers, social researchers, students, and a strong, paid advisory board of impacted women would ground the framework, methods, analysis, and the organizing that would evolve. As Judy remembers,

"That first meeting was the first time I saw the list of potentially eligible women. They were all the women I had literally just left when I got out of prison a few months before. They were my neighbors, friends, sisters I had left behind, and the possibility of being able to do work that could mean they too could traverse that great divide between inside and out inspired me".

As the work progressed, Kathy Boudin, Judy Clark, Anisah Sabur, Michelle Daniel (Jones), Sharon White-Harrigan, Cheryl Wilkins, Kate Mogulescu, María Elena Torre, and Michelle Fine recognized the need for a project coordinator and social worker to facilitate relationships/collaborations with women still inside, and Melissa Mahabir joined. After their release, Monica Szlekovics and Patrice Smith, each of whom brought expertise from longstanding involvement with advocacy, law, and/or research, as well as experience in prison and/or with domestic violence, also joined SJP, and as cases began to multiply, another lawyer was added to the team, Elizabeth Isaacs.

As we began to dig into what we could learn about the 487 women, it became even more apparent how critical it would be to collaborate with women currently inside to unpack the legal language, to help women make sense of the new law, to link them to legal counsel, and to help women craft "new narratives" that might enable petitions for re-sentencing. We learned that a small study group of five women had formed within the prison at Bedford Hills, analyzing the law and its implications for women inside. We

came to understand that women needed, as Cheryl reminded us, "time to develop a new narrative; that doesn't happen overnight. You could do that in college, or with other women, but you need support." Monica elaborated on this point in her thesis, written while she was a Marymount Manhattan College student in Bedford Hills, calling attention to the long process of crafting a new narrative, one that insists on complexity and rejects binaries of victim and perpetrator, guilt and innocence:

> "I am a woman who is both a victim and offender. I am a woman to whom very few accolades are given . . . I am a woman who is seeking out her own truths . . . I am a woman who is actively trying to stop perpetuating the cycle of violence in her life. I am more than a number, the consequences of my disempowerment. I have a history. I have a voice, and I am not unlike you."—Monica Szlekovics (in Ensler and Doyle 2007, p. 217)

Our work as a collective is to understand, experience, listen to each other, even/especially given our divergent biographies, to better understand the deep and complex entanglements of domestic violence and criminalization. In the (dis)comfort of the SJP collective space, we have had hard conversations about where personal responsibility lies between suffering domestic violence and then causing harm. As one woman offered for our Resource Guide, "I felt like I wasn't innocent, but I wasn't guilty either". We do not slide past the complexity but take seriously how deeply and painfully these dynamics entwine. The point of the DVSJA and our work is not to relinquish accountability but to appreciate complexity, not to offer justification but to create space for explanation. Indeed, some of us would argue that the only path to freedom involves addressing harm and responsibility to those hurt, those left behind, and those with whom we will try to give back and repair. In our collective, brimming with very diverse her-stories, we work in the gray, refusing the seductive clarity of black and white.

Since March 2020, SJP has been meeting in bi-weekly Zoom rooms, documenting the history and unfolding of the DVSJA and analyzing the language and strategy of emergent cases. We have also carved open, powerful, and sometimes delicate conversations about race and gender in re-sentencing; concerns about who is included and excluded in the law; the conditions in the prisons during COVID; the deep and painful continuities of domestic violence and state violence; how region and race affect decisions and the likelihood of re-sentencing; and how race and racism affect the kinds of "evidence" to which women may have access (e.g., hesitancy to call police by women of color fearing for their lives, or the removal of their children, or the failure of medical providers and law enforcement to take seriously the complaints of women of color when they do seek support). SJP members have worked on supporting/collaborating with incarcerated women who are applying and have applied for DVSJA re-sentencing and have begun conversations with women recently released and those denied petitions still inside. We have also devoted time to training judges and defense attorneys, collecting data, connecting with advocates in other states interested in DVSJA replication, and building SJP's network of partners. At our most recent day-long retreat, we also envisioned the continued DVSJA implementation work of writing op-eds, training prosecutors, going into the prisons to meet with women, figuring out how to use the framework of the DVSJA to bring more attention to the moment of arrest, and brainstorming ways the DVSJA itself needs to be improved upon and expanded.

## 5. Our Labor to Date

SJP has evolved a series of specific goals, and the list continues to grow. Initially we worked to identify and gather as much information as possible about survivors in the women's prisons who were potentially eligible for sentencing relief under the DVSJA. Using the DOCCS 487, we curated an expansive database with more than 150 "variables", including name, county, race/ethnicity, sentence, conviction, time served, age, judge, and prosecutor, adding any information evident in legal papers about domestic violence; mentions in the media; whether or not they applied for DVSJA relief, is connected with a

lawyer, has gone through the DVSJA process, has been denied or granted re-sentencing, or is awaiting a decision.

As the database filled with publicly available information, we were also able to gather letters submitted to the New York State Office of Indigent Legal Services from incarcerated women across the state seeking assistance with the DVSJA. We connected the women with lawyers where possible, and we began to understand, from a sample of the letters, the complex emotional entanglements of domestic violence and criminalization; we could hear how and why domestic violence "evidence" was often absent from the court transcripts or was never shared. Dynamics of fear, retaliation, women's trauma, and their silence about domestic abuse lift off these emotionally charged, often hand-written pages. Some women wrote with questions and hope but mostly fears that they would not qualify; a few describe decades of abuse and then advice from public defenders not to introduce the abuse in the trial. One woman explained that her lawyer refused to bring supportive witnesses to testify; one feared revealing the violence as her mother sat in the courtroom; and another offered that she was abused in the military and so sheltered alone in silence. These letters reveal the structural limitations, misogynistic "advice", the social and the psychic silencing, shame and fear that layer over the abuse.

We reviewed the legal, social science, and activist literatures on the entangled gendered and racialized relations of domestic violence and state violence, the retraumatizing effects of prison on survivors, and the absence of mental health supports for women inside (see Davis 1981; Kaba 2021; Richie 1996, 2012; Ritchie 2017). We learned that 80–90% of women in prison report experiencing sexual, partner, or caregiver violence (Richie 1996); that the most severely abused can often be found far from the "general" population of women, tucked away on the mental health unit; that women who attempt suicide are often placed in solitary confinement; that the Family Violence support groups in the New York women's prisons, led by incarcerated women, have been disbanded; and that the federal Prison Rape Elimination Act (PREA), enacted in 2003, has been implemented in ways to maximize sexual surveillance and punishment. We learned that a national study of women in jails found high incidences of mental health problems: 32% of the respondents met the criteria for a severe mental illness in the past 12 months, 53% met criteria for lifetime PTSD, and an overwhelming 82% met lifetime criteria for drug or alcohol abuse or dependence. The study also found that childhood victimization and adult trauma increased the risk of poor mental health, which in turn predicted a greater history of criminal legal involvement (Richie 1996). We learned, again, that violence against girls and women, trauma, abuse, and psychological harassment are a through line in the lives of many girls and women who end up in prison and that prison re-traumatizes, as does parole/supervision post-release.

Early reviews of our growing database confirm racial and regional disparities and reveal how much work has been done—and is yet to be done. As of this writing, a preliminary analysis of the list of the 487 women potentially eligible for re-sentencing revealed:

- A pronounced racial disparity: 54% of the women are Black (as identified by DOCCS) as opposed to 38% in the general population of women in prison;
- Approximately 1/3 are serving indeterminate life sentences;
- Over 25% of cases involve a family member other than an intimate partner/former intimate partner;
- More than half are working with lawyers on re-sentencing applications; and
- Thirteen survivors have been re-sentenced under the DVSJA (along with three men).

In addition, the clinic at Brooklyn Law School has filed six re-sentencing applications and is working to prepare several others. Of the six filed, one petition was denied, three were granted in full, one was granted in part but denied in part, and one is currently pending. From the first 16 re-sentencings in the state, prison time for these survivors has been reduced significantly: by a total of 380 months, or over 31 years. Because many of the new sentences are less than the time already served, it is worth noting that if the DVSJA adjusted sentences had been imposed originally, survivors would have avoided

1398 months of prison (116 years!). Our beginnings represent an impressive amount of reclaimed time and a sobering reminder of the long sentences incarcerated survivors face.

Beyond creating a database, connecting women to legal assistance, contacting women who may be eligible for re-sentencing who had not applied through letters, and responding to questions, we quickly came to appreciate the need to support survivors navigating the DVSJA process—whether their applications were successful or not and, after they are "out", facing difficult circumstances as they return "home". These challenges not only include COVID-19 but also, at times, the precarity of returning to the same neighborhoods where abuse took place and/or friends/relatives of the person who abused them still reside. As women have been released, they remind us of the continued control and surveillance they experience under parole, housing shortages, employment barriers, and difficulties that can come with adjusting to life on the outside after years of incarceration. At least one woman who was released under the DVSJA was then deported and separated from her children. We feel a responsibility to draw on our networks to try to enhance community support for survivors returning home from prison.

The more women share their experiences with us, the more we realize how essential it is to build capacity and develop resources for attorneys working with survivors on DVSJA applications and re-sentencing cases; educate judges about the DVSJA and issues surrounding the criminalization of survivors; and open conversations with prosecutors around trauma, domestic violence, and DVSJA cases.

Finally, we have been moved to watch how the story of the DVSJA travels. We have started to share DVSJA-related information across the state and country, and even internationally, with advocates interested in adopting similar legislation. SJP has facilitated trainings with judges and attorneys and hosted meetings with journalists and reporters and activists and allied lawyers, including organizers across the country interested in how to advance similar legislation for survivors in other states.

## 6. Building and Sustaining Our Research-Action Collective in a Long History of Survivor-Based Movement Building

We turn now to the heart of this article—how we work as a collective across higher education, community organization, and movement activism to co-produce knowledge through an intersectional critical race and feminist lens. You have heard how we built SJP: each of us (see short bios at end) brings distinct gifts, experiences, lines of analysis, things we are certain about, and deep uncertainties. Together we share and have cultivated commitments and accountabilities that we try to make explicit in this article. We have grown a critical praxis, with entangled threads of legal work, research, organizing, popular and professional education, laughter, joy, despair, and more joy. We are committed to epistemic justice, privileging the wisdom of survivors. We work in tender solidarity, across generation/experience/race/ethnicity/forms of struggle, and we hold ourselves accountable to women still inside, women recently released, and survivors who may in the future face the criminal punishment system.

In many ways our work embodies and complicates the commitments explicated by Gordon da Cruz (2017) in her essay "Critical Community-Engaged Scholarship". As a collective, we nurture critically conscious knowledge on three fronts: within SJP; in ongoing conversations with women still in prison and those recently released; and in trainings/networks/strategic alliances with advocates, allies, and professionals who need to understand/be educated about the deep and entangled relationship of domestic violence and criminalization, racism, sexism, and state violence. We try always to engage thoughtfully, ethically, and strategically to unearth and appreciate a broad range of "expertise" nourished and informed by experience, oppression, academic and professional training, resistance, survival, organizing, and movement work. We have built a space where these knowledges and sometimes hard truths can flourish, where our complexities shine, where our gifts are shared—poetry, art, organizing, teaching—and where new insights and incites emerge, even if, at times, tensions flare. Our work is "assets-based", but that does not

mean that we ignore the scar tissue, the cumulative wounds of state and intimate violence, the desires for community, and the ruptures of mistrust, the yearning for a life that might be otherwise, the need for us all to understand, and so much more. From our individual and collective lives in prison, in the university, from movement work, and from sitting with our mothers/aunties/sisters/grandmothers/chosen families, we know that women's knowledge grows fiercely when we share intimacies, in both quiet and public ways; only then do our stories saturated in privacy/shame/trauma reveal structural dynamics. We know that the journey to justice is long and hard, that victory is delicious, and that there is precious knowledge produced in struggle.

We understand our work as movement-based inquiry, a moment in a long her-story of survivor-led struggle. We also place our work in the legacy of critical community-engaged scholarship, as Cruz argues, with a strong feminist and participatory twist and held by collective complex solidarities of critical participatory action research (Ayala et al. 2018; Fine 2017; Fine and Torre 2021). That is, a central commitment of SJP is epistemic justice; we bring various forms of expertise/wisdom together to generate even more radically vibrant wisdom. "No research about us without us" is foundational (Fine and Torre 2021), and so together we form what María Elena Torre calls a "participatory contact zone" (Fine and Torre 2021; Torre 2009; Torre and Ayala 2009), a collective of differently positioned co-researchers/activists/lawyers who bring distinct experiences, wisdoms, knowledges, questions, and ambivalences to the work. As María Elena Torre describes (Torre 2021), "our different histories and experiences are essential to the power of our work—we not only recognize, acknowledge and seek out their meanings, we use our differences to sharpen our collective analyses. We ask questions of and push each other, challenging what we think we know, so that we can arrive to new, more complicated and nuanced understandings."

In this way, we ground our work—legal, research, resource building, organizing, advocacy, and support to survivors going through the DVSJA process—in the knowledge held by the women most impacted by domestic violence and the carceral state. We bring a strong intersectional lens attentive to how racism, classism, and sexism intersect in the criminal legal system, and we trust that, across our various lines of vision, new knowledges—critical, feminist, and multi-scalar—and new forms of action will emerge. Our deep commitment to solidarity enables us to engage difficult dialogues across networks and work at multiple levels: documenting the impact of state law, working across distinct regions in the state, training judges and defense attorneys, meeting with social workers interested in supporting women and their DVSJA applications, collaborating with women on the inside and advocacy groups on the outside, and building cases for individuals in prison.

In the process, we model for our respective universities, Brooklyn Law School, The Graduate Center at CUNY, and the Center for Justice at Columbia University, the power of public-facing scholarship and action; the obligation of universities—IRBs, budget offices, journals, media relations, and those in charge of "space"—to engage in research and action with community, as part of struggles for racial and gender justice, recognizing and validating knowledges borne in struggle and in hell, on Zoom and in community.

We are as committed to co-producing "community engaged scholarship" as we are to engaging action, organizing, policy change, and popular education. We mobilize around what Gramsci, and more recently Akbar et al. (2021) writing on movement law, has called a non-reformist reform: a demand for justice rising from communities in struggle, an incremental element of a much more radical vision of racial and gender justice: decarceration and the abolition of violence against women.

We turn now to some of the sticky questions of power, organizing, epistemic justice, and process—within our collective; with women still inside prison or recently released; and with allies/networks/other states/policy makers. We frame our sections around questions we have been asked, so that other community-based research/action collaboratives may see yourselves—or shared questions—in our praxis.



### 7. Nourishing a Collective of Epistemic Justice and Love

Our research collective commits to epistemic justice, refusing to be extractive and rejecting the notion that only people outside prison, or in the academy, have expertise. This renders the inquiry and the praxis more valid, better able to feed organizing and policy making. Research on incarceration—like any social issue—must be grounded in the lived experience, wisdom, understandings, and desires of those most impacted and in difficult dialogues across communities, struggles, contexts, and even lines of power. Our approach echoes the writing of Hannah Arendt (1957) decades ago in *The Human Condition*, where she argued for dialogue as prerequisite to thinking, where she worried about the "banality of evil" in the absence of critical thought, where she advocated "la vita activa"—the braiding of labor, work, and action—as essential to political agency.

This commitment to critical participation, epistemic justice, and to research led by those most impacted has long been foundational for our collaborative projects. We all bring distinct forms of expertise and experience—history as organizers and as teachers—and yet we share a recognition that the path for the work must be led by those most impacted. As noted by Kathy Boudin (Boudin 2021), who was involved in the 1985 Bedford hearing and the 2001 Changing Minds evaluation of college in prison and now teaches social work and co-directs the Center for Justice at Columbia University, addressing a range of projects that bridge the university and movements for justice and freedom "beyond the bars":

> "It is crucial that this project is rooted in the perspectives of those most impacted; when given a chance, those impacted have the understanding that can lead to solutions, movements, education, law. It's the people who have experienced that ... sometimes people are so vulnerable they can't initially lead the process to get it going, but if our process builds on their strengths, they will be the leaders, their drive; our responsibility has been to create a process like that at every level. Over and over, I have seen that happen—it's up to the people who are not that vulnerable to create a space for those who are most impacted to become leaders and express themselves".

This connection between activism and voice—the core of those most impacted and the conditions in which critical consciousness erupts in a community of trust and love—can be heard when Anisah dates "finding my voice" to her first Advocacy Day with the Coalition for Women Prisoners, remembering her travels to Albany to meet with legislators and advocate for the rights of incarcerated women. Showing us a beautiful photo of a younger self, Anisah explained this as a moment of "being and becoming an activist, once I was released, allowed me to find my voice".

Sharon, too, narrates her moment of critical insight and incitement—linking domestic violence and criminalization of surviving—back to work in the cosmetology unit at Bedford, an experience that became the match that has lit her ongoing commitment to advocacy.

> "Maybe six years after the Domestic Violence Hearings, I got to Bedford. It was in cosmetology, I did hair. As I listened, I realized maybe three quarters of the women, and some of the Correction Officers. I heard story after story. The details were different, but they were broken, smashed in a relationship with a man. I did time, I did hair, I listened to the women, I went to college, and I learned from my peers. The programs were not enriching but speaking to peers, hearing their stories. The common storyline was relationships, being violated in a relationship. So, I learned the language of trauma—"

Sharon spent her years at Bedford, she explains, trying to understand the grief, pain, and fear that accompany domestic violence.

Critical consciousness emerges through deep conversations with other women behind bars, cutting hair, listening to the echoes of trauma spilling out of women's mouths, participating in peer-led alternatives to violence programs, and attending college classes behind bars.

"When my anger kicked in and I went into [another] mode. In college [in prison] I learned to put names to what I was feeling, what I was experiencing. I was shifting".

Reflecting on 11 years at Bedford and one at Rikers Island Jail, Sharon recalls,

"Really, I feel blessed. Did 11 years, and I never would have come to know myself, know these women if I didn't go to Bedford. I had to go through that to become me, the MSW, working as an advocate for impacted women".

We have learned, again and again, that together, in difficult conversations—whether in cosmetology, in prison college classes, or on Zoom—knowledge and action emerge; radical imagination is provoked.

The organizing to pass the DVSJA created ripples of possibility (even when complicated) for women at Bedford even before it was signed into law, as Judy recalls:

"While I was in, there was an effort to get the law passed. There were women inside who worked to get the law passed and met with Sister Mary Nerney and felt hopeful about that, but over that same period, the prison clamped down on outsiders coming in and shut down volunteer programs. At the same time, the parole board was systematically hitting women with violent crimes, and no one was getting clemency, all of which left women feeling overall isolated and discouraged. There were always some women waiting for the law to pass, knowing it could make a difference, and other women making connections with outside groups. Since the DVSJA passed, some clemencies have been granted and the Parole Board has been challenged to release more long termers, so there is greater hope and activity".

This was echoed by a woman in one of the letters sent to Indigent Legal Services. She explained that she did not see herself as a survivor of domestic violence until she got to prison and participated in the Family Violence program:

"With the help of the Family Violence program offered here, I addressed my fears . . . and became a facilitator for AVP (alternatives to violence programs)".

In the community of women, sharing stories of violence, a new grammar of violence and justice, new identities and commitments, develop.

Patrice, one of the first survivors to be re-sentenced and released under the DVSJA, points to moments in the last few years of her time inside when language, identities, and consciousness were provoked by organizing with other women in prison, and then out, around this law:

"I was so traumatized by being vilified in the public eye when I was only 16 that it took 20 years for me to re-see myself as worthy of a second chance. It was very hard for me to join a public campaign on my own behalf, it took the support of women who believed in me and the law to do it. Right when the DVSJA was passed, I was shipped off to Albion, a medium-security prison up near the Canadian border, and it took a while for me to hear about it. At first, I wasn't even sure the law applied to me, and I was so discouraged from having lost a number of appeals that I was afraid of feeling false hope. Nobody at Albion knew about the DVSJA, so once they heard my application was accepted for a hearing, I became the go-to person for everyone's questions. I knew my case might set a precedent, but I didn't realize the impact it would have. It's significant that the judge who originally sentenced me to 25-to-life publicly stated that our criminal justice system allows for mercy and for reflection when the defendant herself is a victim.

As soon as I was released, I joined SJP, and it empowered me. When I say I'm a survivor now, with this group, I feel more powerful. When I speak with Monica and we say we're survivors, I feel strong in that identity, and I feel the group has given me that. In other spaces, I always feel a level of pity, and I want to

say to people, "I don't need your pity; I want your anger", but I don't feel that pity here."

In our conversations, we hear new narratives unfolding. We learn how trauma affects a woman's likelihood of seeing herself as entitled to justice; about women's worries about reopening past wounds during the re-sentencing process; women's ambivalence about exposing their children/families/selves to reliving the trauma in a re-sentencing hearing; reluctance to face the same judge or prosecutor 10, 15, 20 years later; the traumas reignited by prison, by parole, or by living in a shelter during COVID-19 with an adolescent daughter.

*Dynamics Within: Bold Moments of Solidarity and Tensions*

In our research meetings, we analyze the decisions of judges and the words of prosecutors who oppose re-sentencing as well as those who support it. We have conversations with women who have been through the DVSJA process—those denied and those successful. We hear, often, how grueling it is to relive the moment and what came before, and to be scrutinized again, by the same judge and prosecutor. We are enraged when we hear prosecutors question women's morality or motives. We engage hard conversations about gender and race and how these dynamics operate among us and our many differences. We know that prosecutors worry that the "floodgates" may open as more women learn about the law. We know the DVSJA has flaws and yet is powerful as an opening tool toward racial and gender justice.

We often discuss language and the binaries that limit our sense of justice.

After our last retreat, Monica sent us a copy of her thesis for Marymount College, at Judy's prompting. The thesis opens with a quote from Francois Taillandier's *Breath*: "There exists inside each of us a hidden, unknown being who speaks a foreign language and with whom, sooner or later, we have to strike up a conversation". Monica begins the thesis:

"In 2004, I was serving my seventh year of a fifty to life sentence at Bedford Hills Correctional Facility, a maximum-security facility for women in New York State. It was a year I would come to remember as the year of self-discovery and healing. Everywhere I turned there were reminders of the pain and the anger I had long ago buried. I heard my voice in the stories told, stories of mental illness, alcoholism, neglect, rape, drug abuse, domestic violence, violence, all markers on a timeline that led to my incarceration.

My own anger, pain, and suffering were pushed aside the moment I was arrested and left to defend myself against the accusation that I was a cold-blooded killer. As I internalized my pain, I simultaneously took in everything that was said about me and I began to wear the label of perpetrator, burying the victim deep inside me. At the time I had no identity or self-esteem to carry me through. I was what others made me out to be, and that person was culpable for the crimes that took place. By only allowing myself to connect to the shame and the guilt, I felt, I had effectively disconnected myself from the victim in me who was abused and terrified. As a result, I remained emotionally stagnant for years.

A year after I arrived at Bedford, I started taking college courses, and in June of 2005, I received my associates degree from Marymount Manhattan College. In the preceding year, I had begun to identify myself as not only a perpetrator, but a victim, too. The process of seeking out my own truths, in a non-therapeutic setting, enabled the victim and perpetrator in me to co-exist . . . It was during this phase of my life that I was introduced to restorative justice. That part of me that identified as a perpetrator desired forgiveness; that part of me that identified as a victim sought healing. After reading A Human Being Died that Night: A South African Story of Forgiveness, by Pumla Gobodo-Madikizela, I knew that both forgiveness and healing were possible. Hence, my interest in restorative justice philosophy was born and academia was the platform I used to explore it." (Szlekovics 2009, p. 1)

Monica pushes us to continually disrupt the binaries that are so easy to slip into even as they so dangerously constrain us. Our collective works to open small quiet conversations among us or with others about our pain/shame/desires, conversations that often feel forbidden yet are the very spaces where critical knowledge is born. This is not an easy process. Our rich and vibrant diversity also holds pain and tenderness from individual and collective traumas of enduring structural violence and oppression that each of us is differently positioned within. As we collectively venture into deep and honest inquiry, centering and engaging the wisdom that our experiences have taught us, new solidarities emerge, and/but often, ruptures flare. We call these moments "choques"—inspired by Gloria Anzaldúa's (1987) understanding that even the most pain-filled clashes hold powerful insight, learning, and growth. We take turns holding each other through these moments and have watched as they sharpen our analysis.

One of our choques erupted in a meeting after the first person—a man—was released after a DVSJA re-sentencing. The announcement was shared over email and initial responses of celebration were quickly met with searing disappointment and anger. We had just lived through the uprisings of 2020 in response to the police murders of George Floyd and Breonna Taylor, demanding an end to centuries of racial injustice. The fevers caused by interlocking systems of racist violence, sexual violence, class stratification, and "the war on drugs" were all too familiar in our communities and in our bodies, and each of us in different ways had had enough. There was no room for the joy of success expressed in some of the responses. As Sharon explains:

> "This law was struggled, fought, advocated by and for women of color . . . we didn't do this to get Black men or white women out first, or even primarily. When is it our turn? When do Black women get justice?"

Sharon's passion, honesty, and pain opened space for others' frustration and ambivalence. The exchange—initially all over email—was tense as folks joined in, sharing the anger, contradicting, justifying positions, trying to clarify where they stood. At our meeting the following day, we dove head first into what happened (some of us probably wishing we could just brush it away, some of us feeling as though facing it was the only path forward), unfurling long conversations and silences about how race and racism, class and classism, gender and sexism operate/collude/perpetuate gender-based violence; how we saw and experienced this in our lives and in our research; how we knew it mattered what region in the state a woman lived, what race she was, if she had money, if drugs were involved, because each of these things impacts who calls the police, who is believed, who is deemed "worthy", who is supported or vilified, who fears the police as much as her abuser, who rightly worries her children might be taken.

The raw emotions that were shared forced us to see ongoing dynamics of racial violence and the ways we have been sold siloed understandings of oppressions that are actually entangled and reliant on each other, but again, this was, and is, not easy work. As Monica reflected afterwards,

> "I felt like I had to legitimate myself as a survivor, to legitimate the harm that I experienced. We started while there was real anger of the Black Lives Matter movement that was part of our work too, but in some of our conversations I felt like I was no longer seen. It was a struggle for me to claim my abuse and what happened to me. In the frustrations that surrounded the release of a Black man as the first after the law passed, someone said to me that the reason I was released was because I was white. That my sentence, my release, was all because I was white. . . . I think when we look at the data we're going to see that white women upstate are getting much longer sentences than Black women downstate".

We share this choque to demonstrate that, while painful, and even enraging, the difficult clashes that can emerge when each of us speak our truths can spark new depths of understanding. Our collective is anchored and driven by a shared commitment to supporting survivors and ending gender-based violence. Our commitment fuels our col-

lective research and also the simultaneously fragile and powerful sisterhood among us. It is this politic and sense of ethics that enable us to enter into and hold these conversations with raw honesty and care. In the language of today, we do not just "see each other" or "hear each other"; we do both, and, perhaps more importantly, we engage each other. It is this engagement that opens up new lenses into the work, provoking new questions/complications/intersections that emerge as inquiry points. The challenges lifted by Sharon and Monica and our collective conversation encouraged a deep look at the racial and class inequalities in our database, analyzing for familiar patterns and those that might be unexpected. While we are highlighting Sharon and Monica in this experience, all of us were forced to confront and rethink, even as differing amounts of small paper cuts lingered on our souls long after the original exchange.

## 8. Accountable to Those Left Behind and Those Recently Released

It seems obvious now: a law is only as good as the mobilization and movements that precede and follow. We knew we needed to be in conversation with women inside prison if we were to animate the possibilities of the DVSJA for those who were potentially eligible. The legislative victory was the beginning/middle but not the end of organizing, but we were mid-COVID-19; women still imprisoned needed support to understand the law, to see where it might be relevant to their case and deserved a community to help them self-author new narratives, "find my voice", "reject the binaries of perpetrator and victim", and "translate my anger into activism", as Sharon, Anisah, and Monica have described. This process is not automatic because a law is passed or a door is opened. Cheryl repeatedly pointed out,

> "It takes years for women to create a new narrative. If they come in believing themselves to be criminals, it takes time, support, and emotional energy to feel entitled to tell a different story about their crime and their identity."

SJP decided—even in the middle of COVID-19—to reach out to women inside, to offer support. Through the Coalition for Women Prisoners, we wrote to incarcerated women that we thought might be DVSJA-eligible but had not applied, inviting them to reach out if they wanted more information or support for an application. After introducing herself and SJP, Anisah wrote:

> "We are reaching out because we want women on the inside to know they may be eligible for this re-sentencing, and we want to offer support through the process to those who apply. So many of us have experienced abuse (physical, emotional, psychological, financial, spiritual, and sexual) from our family members and/or partners, and it is a systemic problem. These traumatic experiences often leave behind shame, confusion, anger, and fear from what we have experienced and what we have done. Please know that you are not alone. We stand with you, we fight with you, and that we hope we are able to provide support."

Shortly thereafter, Melissa, Judy, Anisah, and Monica (along with Sharon Richardson, Executive Director of Re-entry Rocks, one of SJP's organizational partners) formed a subcommittee to create a Resource Guide for survivors applying for DVSJA sentencing/re-sentencing (www.sjpny.org, accessed on 11 December 2021). The goal of the guide is to demystify the language of the law, which may feel limiting or intimidating, and to provide support and insight into the process as survivors move through the different stages. The resource guide explores practical components of the application process, including identifying a support system, gathering evidence, and what to expect at each step. We also delve into aspects of the process that we anticipate may be emotionally challenging: waiting for decisions; processing and sharing a narrative of abuse that may include new information or a part of someone's experience that is different from what was originally shared with the court; admitting involvement in the offense, which may feel scary, because the criminal legal system is so hostile; and facing the same judge and potentially the same

prosecutor from the original sentencing. We used our own lived experiences and the insights from women who were incarcerated when deciding what to include.

Once a draft was completed, Melissa, Judy, Monica, Anisah, and Sharon reached out through letters, phone calls, and Zoom sessions, to women inside, and those newly out, to ask if they would collaborate on the guide. Five women inside and five women who have been re-sentenced and released through the DVSJA provided feedback. The conversations helped us understand the law's shortcomings, the degree of retraumatization produced at a re-sentencing hearing, what women need when they are released, and just how much they want to make sure the law will be successful for other survivors. Anisah commented on the process of reaching back to women inside and those recently released:

> "To hear women talk about this, even to hear one woman still inside, who called into our research group—it was heart wrenching but so useful to build a guide to help them grow a new narrative—but they fear being accused of "lying" now that their story is so different, so much fuller than the first time around".

Through the entire process of creating the Resource Guide, Myeshia Hawkins-Taylor, a woman who applied for DVSJA re-sentencing but was denied, has been guiding our work. From prison, she has been communicating with us on the concerns women have inside about the process and what applying was like for her. While Myeshia knew that she could be denied, she decided to apply, because, as she said, "I want things to change, and I want my story to help others".

This commitment to supporting small, quiet, intimate conversations and building critical consciousness within the prison, in communities, and even with judges, prosecutors, journalists, and with women recently released, is a crucial and often ignored element of participatory praxis.

As Melissa argued:

> "It's important for us to connect with women inside to build the Resource Guide; we don't know their experiences in the current moment—fears around applying and why they may not apply. We need to hear that, and the language we use in the guide matters".

> Anisah joined in, reflecting on her early role in advocating for the DVSJA but acknowledging that it has been a while and that listening to women talk about current conditions was "heart-breaking".

> "Although I played a major role in advocating for this law, I didn't know how it would affect women who would apply back and get released. We didn't think, in the law, about the re-entry impact. It was heart wrenching. We spoke to a number of women for whom the re-entry experience was so horrifying they would rather continue their sentence versus getting out sooner and not getting any support. So, the new question is how do we support women AFTER they are released? How can we put resources in place after they come home? It's one thing wanting to get someone released but if they are coming back to the same abusive community, same home … If they were convicted of killing a loved one, what are the family dynamics on the outside? Supportive or not. It was a whole lot that I learned as someone who experienced incarceration but not as a survivor. To hear someone talk about how the DV and the incarceration and then supervision [parole] all reinforce the trauma.

> Some of them who came out—they were horrified by the systems they had to navigate. They had to deal with their abusers' families who were not supportive of them getting out; their kids; the shelters during COVID. Some women who were inside who thought it made no sense to apply for the DVSJA—they knew they had a lack of support, going back to the same judge, not having the supports that were needed to go through the process. Some women feared going back on trial, gathering up information, being exposed to the same DA, presenting before the same judge. But for others, they are excited. Now they want to tell their own story in their own language, with a new narrative—not the one the media/judges have told about them."

Kathy leapt in on the question of re-entry, recognizing that DVSJA offers some distinct challenges but that re-entry is a broken system for all.

"Re-entry process is just like re-entry for all; more than 60% go back within three years. The re-entry situation is terrible—housing/the family doesn't want to see you around again; if you killed somebody you are going back to the same neighborhood—it's not just re-entry for DVSJA. There are particular things about DVSJA that stand out—but when coming home you are excited, and then it doesn't work out as you imagined. Can't single out DVSJA, and yet there are specifics.

The trauma of working through one's life and what led one to do what one did . . . it's really, really hard. Just need a lot of help and support. The resource guide is an attempt to accompany women through this process, with supports for women inside. Very, very hard. Then to face back up to the same judge—our understanding comes from people dealing with this."

Melissa summarized our learnings, from conversations with women inside and out:

"I want to add four more things I think we have learned from collaborating with women inside and those recently out.

Most exciting, we were struck by how much the women wanted to be a part of this—the stories were pouring out—and the women know they are contributing to our larger work, and they want to be a part of this movement. One woman said I knew I would be denied. I didn't have high hopes but wanted to contribute my story, my knowledge; they were so invested in contributing.

Second issue involves the pain and struggle of re-entry. Free doesn't mean free. We are learning what we need to do to support women upon release.

Third, there are painful lessons about why people won't apply or why women are scared to apply. The theme is the trauma; going in front of the same judge, remembering how difficult, how the DA treated them; the judge "made me out to be a monster", and I can't go there again. Some women would tell us they would not have applied for DVSJA when first arrested because of fear of an abusive partner, or fear of what their partner would do to family, or they loved their partner. So that raises questions about the likelihood that women will apply for DVSJA at the time of their sentencing. The law was intended to help women at the point of arrest—but the women are now saying because of trauma they wouldn't, and the law is written in a way so that if she didn't apply at the moment of arrest, she can't go back.

Finally, we learned how difficult it is for women to understand the law. All they have is the legal language. One woman said—they are going to determine if my abuse was substantial enough; how are they measuring that? I need to know. So, in the guide we tried to help make sense of it . . . we are still learning but want to help women who might be confused . . . "

Judy added the important observations:

"While it is true that we spoke with women who face daunting conditions once out, we also spoke with women who are thriving, in part because they had support from family, re-entry organizations, college, and other connections they made while inside. Not only are they eager to help in our efforts, they are also doing work that provides support for others. Similarly, while women spoke about the trauma of having to go back into their past experiences of abuse, several also felt that they emerged from their struggles stronger, because they were able to voice their truths. Several also talked about being heartened by the successes of others. As is our tradition at Bedford, women feel that the rising of one is the rising of all."

Some of us are also haunted and frustrated by the women we know who are excluded from seeking relief from DVSJA, even though their stories are so similar. Women who have life without parole sentences, women with first-degree murder convictions, women serving less than eight years, women whose abuse happened during childhood, before the instant offense and, therefore, discounted. All deserving. All excluded.

## 9. The Tendrils of Our Work: Building Movement-Based Solidarities across County, State, and Transnational Borders

In this final section, we catalogue a series of what we call "tendrils" of our work: strategic interventions with judges and prosecutors, with advocates in other states, and with departments and institutes across a variety of higher education institutions to raise consciousness about the links between domestic violence and criminalization; to inject feminist and racial justice sensibilities into conversations about the carceral state; to detail the opportunities and limits of the DVSJA; to refuse the binary of "victim" and "perpetrator".

We individually and collectively spin tendrils and bring those back to our collective gatherings for collaboration, insights, cautions. We are building relationships across disciplines, networks across states and internationally, with judges/prosecutors/journalists, and engaging with university research collectives. With limited room, we offer a few images:

- Judicial Trainings: Sharon, as a Reverend, social worker, and the Executive Director of the Women's Criminal Justice Association and #BeyondRosie's struggle, has trained judges about the DVSJA, focused on the words judges say to women facing prosecution and punishment—how they might frame, to the jurors, a woman as simultaneously a "victim" and "defendant". In her own words, she lays out the elements of the law, the elements of trauma, and the significance of language. To judges, she is clear:
- "Language is important! Perception is key. You have an opportunity to potentially make a difference in how women are viewed in the legal system. When we talk about equality, justice, and fair chances—we need to think about how women are presented and represented in and out of the court system. Every woman, especially Black and Brown women, has a traumatic history. When in the courtroom, they should be addressed as survivors, rather than defendants".
- Mobilizing with sister campaigns in other states: We have been invited to collaborate with activists, lawyers, and emergent coalitions in Connecticut, Oregon, and California. In addition, Michelle (Daniel Jones), has been working with activists in Indiana on various legislative efforts to eliminate barriers to incarcerated women. When she learned about the DVSJA in New York, she became interested in trying to build a movement to support a sister law or legal intervention in Indiana. As a member of the Marion County Re-entry Coalition of Indianapolis, she learned about the Conviction Integrity Unit (CIU) based in Indianapolis, a department of the county prosecutor's office tasked with identifying and remedying false convictions. We will be meeting with them soon.
- Decolonizing academic and community-based conversations about epistemic justice, participatory and public-facing scholarship: When offered opportunities to deliver a keynote/panel/present at a conference or university, Michelle Fine often asks to convert these into opportunities for our whole group to present, requesting funding for those who are not already employed by a university. We have presented, to date, at the "All In" conference sponsored by URBAN at UC Santa Cruz, with a university–community coalition of "inside/out" researchers and activists at the University of South Africa in Pretoria, with students in a new PhD program in Community-Based Activist Leadership at the College of Staten Island, and we are slated to present at the Yale University Psychology Department Clinical Series. These talks are designed not only to present on our commitments and "findings" but to model the obligation of universities to engage in public-facing scholarship, movement-based work, and interdisciplinary praxis with communities in struggle.

- Speaking truth to power in Congress: In the early summer of 2021, Patrice testified before a special panel on the criminalization of sex trafficking survivors in a Congressional listening session. This symposium was led by those with lengthy prison sentences for offenses committed against those who had sexually abused, raped, or trafficked them as children.
- Moving ideas and campaigns through mainstream and social media: Early in our process, Kate, Sharon, and Michelle Fine appeared on the Ronnie Eldridge cable TV show to share our work at that stage. Ronnie, the host, had been one of the key organizers of the early Bedford Hills hearings. Kate presented on the law, Sharon on the organizing before and since, particularly the struggle to close Rikers, and the Rose M. Singer Center, the women's jail at Rikers. Within a matter of weeks, the painting below (Figure 2) arrived at Sharon's address, painted by a man incarcerated upstate who watched the show and was moved to portray his tears, his commitments, and his political imagination.

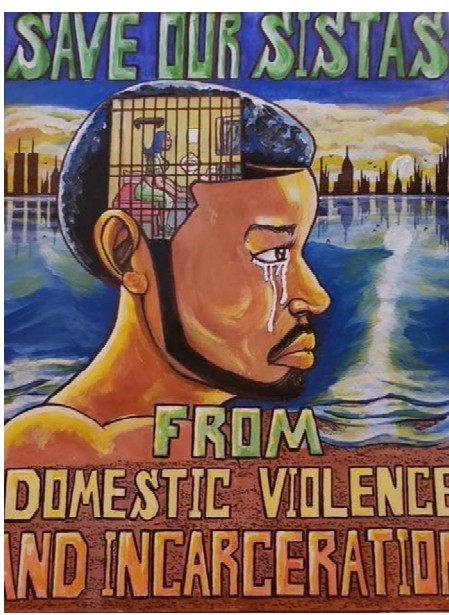

**Figure 2.** Artwork by Bruce Bryant. Sent as a gift from a men's facility upstate to Sharon White-Harrigan after she appeared on Cable television to discuss the DVSJA.

Our work travels, by design and by a kind of radical empathy that moves across screens and people, into Congressional hearings, judicial trainings, college classrooms, organizing spaces, and through barbed wire, in ways sometimes predictable and often beautifully shocking.

## 10. Reflections

When we began this work, we knew that we were committed to critical race, feminist, and participatory methods; that the work should be rooted in the perspectives of those most impacted; that we would have to work hard—through COVID-19—to be in relation with women still inside. We knew that the law was potentially powerful and yet flawed. We knew that we would have to create small learning communities—among us, with women inside and recently released, and with attorneys/organizers/judges/journalists and activists elsewhere. Grounded in the writings of Gordon da Cruz (2017), we reflect here on how our work takes seriously feminist, critical, and participatory praxis; how we embraced, embodied, and enacted deep relational ethics and an assets-based frame; what we learned; where we stumbled; what we now know about the rhythms, ethics, and strategic decisions of feminist praxis grounded in law/inquiry/organizing. We offer essential questions that perhaps any university–community research collective should address.

Who should be included in the research collective? We have evolved into an interdisciplinary collective, intentionally "diverse" and care-fully attentive to questions of power, intersectionality, multiplicity, joy, and struggle, paying for the knowledge, time, and labor of those not affiliated with universities, attentive to how fractures—or "choques"—might erupt. Research collectives that are movement-based must address complex questions of deep and radical participation: who, how, and under what conditions—and then contend with questions of power, i.e., payment for time/labor, travel, credit, academy history of extraction and exploitation, visibility of issues, and vulnerabilities of collective members. As Fine and Torre (2021) and Kapoor and Jordan (2019) have argued, community-based research collectives, particularly those rooted in movements, must center the perspectives of those most impacted, animate the lines of analysis that derive from impacted people and communities, and be intentional about to whom our work is accountable and for whom our work is written.

How can the group address, in an ongoing way, critical questions of power differentials and at the same time nourish a sense of solidarity within the research collective? Turning to the production of knowledge and action—and even writing this paper—there are many questions about how to assure and enact full participation in the writing/organizing/policy work; how we recognize and make use of our very distinct forms of experience and expertise; and how we refuse to fetishize those most impacted. When we are at rallies or speaking to legislators, who speaks for the group? When we write for a journal such as this, who takes the lead? When we approach funders, what combination of us shows up (see Fine et al. (2003, 2020) *Changing Minds*, for more detail)?

For instance, if you look back on this paper, you might notice who is "quoted" and who writes in passive voice as the voice-over? Who is cited from other publications, and who sees this paper as a chance to "find my voice"? These "differences" are gifts, and yet they dance awkwardly at the edges of our work. Our distinct gifts/experiences/forms of expertise blend among us, but when we engage with external audiences, microaggressions can erupt. For instance, when we give talks, an audience member may pose a question that pries a bit too deeply into a co-researcher's life—a question that they would not address to Kate, Melissa, Elizabeth, Michelle, or Maria about our experiences with violence, crime, or abuse. As a research collective, anchored by survivors, perhaps we have entrapped audience members into this dynamic. That is, in SJP, we always make clear that we honor the experience/expertise/leadership of survivors of DV and prison, and yet when audience members interrogate the roots of that expertise, it can feel like a transgression. These dynamics are laced throughout critical participatory projects, and SJP is no different.

Within SJP we usually—but not always—choose to dive into rather than banish such delicate issues, sometimes as a group, sometimes in smaller subgroups off-line. Sometimes the issues are resolved among us, and sometimes wounds linger. So, too, in our relations with women inside, we choose to strategize explicit and complex ways to communicate with women still inside and those recently released, and yet these entanglements also are fraught and complicated by need and circumstance and our knowledge of how hard it is to "come home"—especially now, especially during COVID-19, especially to a family fraught with violence. Third, as we have noted throughout, we have always collaborated with those outside SJP who might be allies—activists, policy makers, progressive lawyers, social workers—but increasingly we also realize that we must engage with those who desperately "don't get it", who need to be educated about DV and criminalization.

Given the deeply unstable grounds of political life, how can a research collective, rooted in feminist and anti-racist commitments, stay laser-focused on the key campaign and remain attentive/empathic to the emergent needs of those most impacted? The goals of our collective have been both clear and directed, as well as nimble and flexible, bending with time and context. We were organized to address the implementation of the DVSJA, and then COVID-19 hit. We felt an obligation to gather the names of women that we believed should be prioritized for release based on medical vulnerability and submitted them to the Governor. As time went on, rooted in community and movements,

attached to women inside, those recently released and those with major advocacy roles, we came to see the ways in which SJP could be available to partners working in allied areas toward gender justice and decarceration; how we might facilitate trainings with judges, speak to Congressional hearings on the criminalization of sex trafficking victims; how we might work with pro bono law firms taking on DVSJA cases, how we might consult with journalists trying to understand the relation of domestic violence and criminalization of survivors; or how racism affects women's sentences and experiences. We believe that SJP is uniquely positioned to teach prosecutors about the meaning and purpose of the DVSJA in a manner that can lead to the decarceration of survivors, to support movements to decriminalize. Because SJP is community-based, new and important questions emerge, because we have ears on the ground. There is a delicate balance between relentless focus on your issue and bridging to new struggles and challenges.

What is the obligation of the university to support public, community-based scholarship and practice? We begin by insisting that our universities recognize this work as public-facing practice and scholarship—not service, not only "ethical" but good public science. Michelle (Fine) argues that "[c]ritical participatory work, grounded in the perspectives of those most impacted and focused on systemic transformation and organizing, cannot be viewed as 'service' in universities but must be recognized as a deep intellectual and ethical project; a praxis rooted in the membranes between university and movements; an obligation of the university to lift up perspectives diminished/silenced/ignored/colonized as legitimate knowledge" (Fine 2021). SJP is no different, and it has rubbed up against many of the "taken for granted givens" of academic life: Who is recognized as a researcher? Who is paid? Who is listed on the IRB? Who gets a university ID? Who writes? Who is credited with SJP? Who is positioned as the "real" survivor of DV and the carceral state? How do race, experience with DV, commitment to the issues, and incarceration influence who constitutes "most impacted"?

We offer this essay as a note on movement-based inquiry, a work-in-progress, draped across the borders of the academy and movements for justice. We sketch here our aspirations, our internal struggles, our desires to engage with those still inside, and our commitments to connect beyond New York and beyond familiar allies. Grounded in feminist, anti-racist, decolonial commitments, sometimes we stumble.

SJP is filled with wisdom, joy, the sweet taste of intersectional sisterhood, and of course, at times, hard moments and existential weight as we consider in ourselves and the women that we work with the magnitude of obstacles facing survivors before, during, and even after incarceration. Positioned as engaged collectively in a powerful and imperfect intervention toward freedom, we are ever aware of the expansive radical transformations necessary for racial and gendered justice. In our work, we are attentive to the large and the small; the ideological, structural, linguistic, racial, gendered, judicial, legislative, the relational, and the intra-psychic, always more complex than anticipated. In this article, we offer up a multi-voiced reflection on our praxis, to document the knowledge, solidarities, and relationships borne in struggle. We write to be in conversation with other struggles, throughout the U.S. and transnationally.

**Author Contributions:** Conceptualization: K.B., J.C., M.F., E.I., M.D.J., M.M., K.M., A.S-M, P.S., M.S., M.T., S.W-H, C.W.; methodology: K.B., J.C., M.F., E.I., M.D.J., M.M., K.M., A.S-M, P.S., M.S., M.T., S.W-H, C.W; formal analysis: K.B., J.C., M.F., E.I., M.D.J., M.M., K.M., A.S-M, P.S., M.S., M.T., S.W-H, C.W.; resources: K.B., J.C., M.F., E.I., M.D.J., M.M., K.M., A.S-M, P.S., M.S., M.T., S.W-H, C.W.; writing—original draft preparation, K.B., J.C., M.F., E.I., M.D.J., M.M., K.M., A.S-M, P.S., M.S., M.T., S.W-H, C.W. writing—review and editing, K.B., J.C., M.F., E.I., M.D.J., M.M., K.M., A.S-M, P.S., M.S., M.T., S.W-H, C.W. All authors have read and agreed to the published version of the manuscript.

**Funding:** This research was funded by The New York Women's Foundation, Vital Projects Fund, Tow Foundation, and Adco Foundation.

**Institutional Review Board Statement:** Not applicable.

**Informed Consent Statement:** Not applicable.

**Data Availability Statement:** Not applicable.

**Conflicts of Interest:** The authors declare no conflict of interest.

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
