# Peer review of "Movement-Based Participatory Inquiry: The Multi-Voiced Story of the Survivors Justice Project"

_socsci, doi:10.3390/socsci11030129_

Round 1

Reviewer 1 Report

This paper constitutes an example of good practices through Action Research, both for the epistemic approach and for the methodology. It is a subject of great interest  and it has been little addressed  by specialized research. My assessment is therefore highly positive. I congratulate the authors for this excellent work and for the very valuable results for future research.

Another aspect to highlight is the adequacy of the epistemological approach, which provides an "extra-plus": reflection on one's own experience, community construction of knowledge, integration of diverse perspectives, conflict as a dynamic element of knowledge ...

This paper makes an extraordinary contribution from a privileged point of view (cfr. Sandra Harding) to understand different aspects of the problem: 1) the violent action of the abused women, 2) the revictimization cuased by the system, and 3) the process of the protagonists for the elaboration of liberating new narratives from their own experience.

I think that it constitutes a very valuable contribution to research on violence against women.

Author Response

we very much appreciate your enthusiastic review; we have added some citations to clarify our conceptual and epistemic commitments

Reviewer 2 Report

  • The paper describes an interesting group undertaking work in an important area of policy and practice with strong commitments and good intentions. However, it is difficult to glean much more than this from the article.
  • The paper does not attempt to engage with extant literature. References are mostly to discussions of research methods and older theoretical works. There is no discussion of the state of the empirical literature. It is therefore impossible for the reader to know how this study is situated in the existing field of knowledge. Does it contribute new knowledge in an existing line of inquiry? Does it dispute or challenge accepted knowledge? Does it raise previously unexplored questions?
  • The aims of the paper require clarification and refinement.
    • Three aims are set out early in the paper:
      • Explore commitment to epistemic justice (including how the group co-produces knowledge ethically, navigates choques, and navigates resource differences across discipline, generations, lived experiences, race, class, and region)
      • Reflect on accountability to women still incarcerated
      • Introduce scenes of delicate solidarities, exploring external relations
    • The first of these could be a purpose for a research paper, but it is not met. There is some discussion describing that choques have been navigated, but we have no sense of how this was achieved. Why is SJP able to navigate these choques? Is it something to do with the composition or governance of the group? Is there a process employed? And how is this similar or different to, and better or worse than, other approaches to conflict resolution (i.e. findings must be situated within the existing literature to generate new knowledge – otherwise, they are mere observations).
    • The second and third aims are not entirely clear. I think the verbs in these sentences – reflect, introduce, explore – are too vague to serve as a research question or to organise a thesis around. As a result, sections that relate to these aims are mainly descriptive, and it is not clear what the audience should take from this. How does it relate to existing knowledge? How could this inform further research, policy, and practice?
  • The paper does not present a description and justification for research methods.
    • Observations and impressions are given throughout. However, no information is provided about the methods of sampling, data collection, or analysis. It is therefore not possible to know as a reader (or as a peer reviewer) whether claims are supported by robust evidence or whether research has been undertaken ethically.
    • The way quotes are presented throughout makes it difficult to disentangle what material is empirical data and what material are the authors’ interpretations of (unpresented) data. Of course, there is a long tradition of researchers taking a participant-observer position, so there is nothing inherently wrong with researchers having a dual role as both participant and observer. However, this needs to be properly explained with the researchers’ positions as insiders managed in both the collection of data and interpretations of findings.
  • To address these issues, I would suggest using a traditional structure for an academic paper to ensure all elements are presented and accessible to the audience: Introduction, review of literature, aims and research questions, methods (design, sample, data collection, analysis, ethics), findings and discussion, conclusions and future direction.

  • Some other general points:
    • There is a tendency throughout the paper to use language that is ambiguous or vague.
      • For example, there a many instances where several terms are provided with a slash between them, e.g. “psychological abuse/coercive control”, “relationships/collaborations”, “privacy/shame/trauma”. It is not clear if the slash in each case means “or”, “and”, or if the terms presented are intended to be interchangeable synonyms.
      • As noted above, some of the aims of the paper are not sufficiently clear.
      • In describing SJP’s work, there is a tendency to provide abstract descriptors without more concrete descriptions. Put another way, the paper presents the conclusions reached by the authors without laying out their evidence and reasoning. For example, at the conclusion of the discussion about how SJP deals with choques, the authors conclude that “We share this choque to demonstrate that, while painful, and even enraging, the difficult clashes that can emerge when each of us speak our truths can spark new depths of understanding” (line 626). However, there is no previous description of who reached a new understanding or how they reached it, and no supporting data to demonstrate this. It is therefore not clear to the reader precisely what it means to reach “new depths of understanding”.
      • In other cases, terminology can be vague, for example, the paper states that SJP “braids” the different elements of its work. The term is evocative and perhaps even poetic, but gives the reader no sense of the structure of the organisation or how the different elements of the work fit together. What are the synergies and tensions between different elements of the work? It is particularly difficult discern the intended meaning of Section 5. For academic (rather than literary) writing, it is better to use precise language, even if it seems unexciting or plain.
    • There is a tendency to make broad statements as if they are truisms, but would actually require considerable justification. E.g. the paper opens with the claim that “a law is only as powerful as the legal advocacy, relentless organizing, and community based inquiry that comes before and comes after”. But there are plenty of laws that are mundane, procedural, and hidden, and that exert insidious power. I would recommend removing these flourishes unless they are material to the thesis of the article and can be supported with evidence and reasoning.
    • It is obviously important to position this work in the intersecting structures of gender, race, and class. In doing so, avoid assuming too much consensus or even knowledge about these issues in the audience. For example, the unique position of incarcerated women of colour may seem self-evident to the authors, and it may therefore seem obvious why this work should be led primarily by and for women of colour. However, this needs to be articulated fully, firstly, to help the reader understand how this paper is positioned in what are often complex and nuanced debates, and secondly, acknowledging the international audience for this journal. Readers (like me) from outside the USA can only have limited and second-hand knowledge of the USA’s unique history and discourses around race, gender, class etc.

Author Response

Clearly this reviewer is expecting a more traditional paper, in that sense, we would disappoint.  i hope such readers can see the value of this form of paper - we added a section with reviewers like #2 in mind.

Reviewer 3 Report

Thank you for the opportunity to review this interesting manuscript.

This piece discusses an important social and criminal justice topic: the administration of justice with perpetrators who are also victims of violence. In this case, the piece discusses legislation (DVSJA) implemented to address this issue and what it takes for researcher/advocates to serve community interests with a suitable community-focused research methodology ('movement-based participatory inquiry').    

Overall, this piece was very well-written and somewhat seamless given there appear to be a number of co-authors contributing to it. The piece reads like a composite of reflections on experiences, praxis and perspectives on the DVSJA in the context of an historical narrative ("a 10 year struggle"). It makes for a compelling statement of challenging epistemic injustice and community healing in a space where voices can remain unheard.

However... and this is where I feel conflicted... as a piece of research I find this manuscript problematic. There is no research question or findings as such. To be fair, implicit in the narratives are evidence of research activity (e.g., user voice; team member commentary) that deserve to be unpacked and presented in their own right and present this work as a programme of research, rather than a monolithic statement (perhaps this has already occurred?). As it stands, this piece - in it's current form - may be better-suited to a publication that favours reflective pieces and/or practice-informed research rather than applied research per se.

The insights are certainly valuable for thinking about what it takes to do meaningful research with marginalised people in the criminal justice system (where the wheels of change are slow) and the message of courage, commitment, and persistence is loud and clear in this work.

On balance, this is a fine piece of work that deserves to be published - and read. But taking into consideration the aims and scope of this journal, may not be a good fit for this publication. 

On balance, this is a fine piece of work that deserves to be published - and read. My recommendation is based on the quality of the work and the relevance of the topic to the focus of the special edition (which I think are high).  

Author Response

we have taken these comments very seriously and addressed within.  We added a section on the shape of the article; we link our work to conjunctural crisis (massey and hall, 2017) to signal that complex, compounded crises need and deserve research collectives that are interdisciplinary and cross-sector; need to think about to whom we are accountable and TO whom we want to speak/influence.  thanks for your thoughtful read and suggestions

Reviewer 4 Report

Thank you for your interesting paper submission, please find my comments below - your work is really interesting and i can really see the power in empowering the voices of those with lived experience here. 

I wonder why you didn't include something around women, or women's justice, in your key words? Not compulsory, just a suggestion.

This is an unusual research paper in many ways as it is written with such a personal style and advocatory tone - however having read the article i entirely understand and commend the need for this approach in this case. With this in mind, i would like to recommend this paper for publication in the journal, pending minor amendments and changes. 

Though i appreciate the tone - the tone does stray into an almost verbal paper at times with phrases like "you will hear" rather than "you will read" or "we will discuss" etc. It is a written, not a spoken paper and thus it would form a more appropriate academic tone to swap this language in your introduction.

Section 2 is good -  You have included a photograph - can i please check that you have the appropriate consent and ethics in place to include this in this forum.

There are several claims in the text that aren't substantiated necessarily with formal evidence however as long as the advocatory nature and tone of this piece is fully acknowledged up front in the introduction, this is not an issue for the style of this piece.

Section six is also a really good and valuable contribution

You do use very large quotes at times and these are, whilst interesting and important to showcase directly from voices of experience, large chunks of text that could be broken down and abbreviated or better integrated, even as figures or case examples at times,  into the wider writing. This is a stylistic suggestion however not a mandatory change.

Line 534 "Choques" heading - formatting needs correction

Sections 8 and 9 are the most beneficial and valuable contribution made by this paper in terms of tangible lessons for other projects to draw on - again there is another image here in section 8- though not a photograph do you have permission to share this in this context.

References are provided and well formatted

Overall - though a somewhat different paper for an academic journal i do believe this article has significant merit and makes a beneficial contribution that deserves publication. 

Author Response

thanks for your thoughtful read; yes we have all permissions; removed choques; narrowed some of the quotes.  this is the tension between inclusion (of a large and varied group) and writing "over" the collective.  thanks for your thoughts - tried to change "you will hear" kinds of framings so it doesn't sound like a speech.  glad you appreciate the sections on what we have learned and how to disseminate the work in collaboration with other movements.

Round 2

Reviewer 2 Report

Whatever the merits of this paper, it does not conform to the requirements of this journal or to accepted conventions of social science. I think all three of the other reviewers and the authors of the paper recognise this. Certainly, Reviewer 3 acknowledges that the paper is not suitable for this journal and the authors have updated the manuscript to state explicitly that it is not written as a research article. Likewise, Reviewer 4 describes the paper as “unusual” and “a somewhat different paper for an academic journal”. Reviewer 4 does suggest that there is good reason for the paper to be written in this way, but these reasons are not set out and are not obvious to me.

Reviewer 1 does not directly indicate that the paper does not conform to the requirements of the journal but, like the other two reviewers, indicates the paper has no empirical evidence by checking the “not applicable” box in response to the question “For empirical research, are the results clearly presented?” Reviewers 3 and 4 go further to indicate not only does the paper have no empirical evidence, they checked the “not applicable” box in response to the question “Are the research design, questions, hypotheses and methods clearly stated?”

However, the paper clearly draws on empirical observations. Large sections of the paper are entirely descriptive. Lines 323 to 342 report quantitative findings from the authors’ own database. There are sections dominated by direct quotes from research participants. Citations are almost never provided for factual claims because they come from the authors’ own observations. It is not that the paper does not contain empirical data; the issue is that the findings are not presented in any clear or structured way and are not differentiated from the authors’ own interpretations and unsupported claims. The paper is empirical. It just fails to meet the basic standards for reporting data. Given that the paper is clearly based on empirical observation, there can be no justification for failing to explain how the data were gathered and analysed. Again, it is not that the question of research design is irrelevant for this article. The issue is that the methods, whatever they were, are not described or justified. This is both a methodological and ethical problem. There is no way for me or the other reviewers to know whether this research was undertaken ethically or what aspects were approved by an ethical review committee.

My objection to this paper is not, as the authors suggest, merely that it takes an untraditional form. The problem is that the paper fails to include any description of the research methods or to delineate between data (e.g. what research participants say), interpretations (e.g. what the authors think the data mean), and the authors’ own unsupported suppositions. (As a side note, I disagree with Reviewer 4 that it is acceptable to mix research findings with “claims in the text that aren't substantiated necessarily with formal evidence” because the paper takes an advocatory tone). One way to ensure methods are adequately described and findings are properly distinguished from interpretations is to follow the standard format for an academic paper (as per the instructions for authors https://www.mdpi.com/journal/socsci/instructions). If the paper is to deviate from this structure, it must surely still contain these basic elements.  

I hope it is not necessary to make the following argument, but just in case, I would like to provide some reasons why it is important for an academic journal, one entitled Social Sciences no less, to insist upon scientific rigour in the material it publishes. There are many forms of knowledge and many ways of gathering knowledge. Epistemological traditions broadly classified as ‘scientific’ may dominate popular and political discourse, but that does not mean that science is the best route to useful knowledge in all cases. But it is the best route we have to certain kinds of knowledge. It is dangerous to confuse one form of knowledge for another. Practitioners of any craft know it is dangerous to apply narrow technical expertise when practical wisdom is required. It is likewise dangerous to confuse personal narrative for science.

If this article is accepted into a peer-reviewed academic journal, it will carry the institutional weight of the academic community. However, it will lack the features of an academic inquiry that are implied by that endorsement. Other academics, policy makers, and the public at large have an expectation that a peer-reviewed article meets certain standards of rigour. The peer-review process is intended to ensure conclusions do not go beyond what the data can support. It is intended to ensure papers are published with enough detail for the broader academic community to continue to scrutinise the methods for gathering and analysing data even after the paper is published. It is intended to ensure research is conducted ethically. It is intended to ensure new research engages with previous research, and that future research can engage meaningfully with past research. 

As it is currently written, this paper does not meet these standards. Publishing works such as this as social science not only asks for readers to misinterpret its intention and rigour but damages the credibility of social scientific disciplines, not to mention the standing of this journal. I do not dispute the other reviewers’ claims that this material should be published somewhere, but it is not social science and it would be wrong to hold it out as such.